# Automated Mass Photometry of Adeno-Associated Virus Vectors from Crude Cell Extracts

**DOI:** 10.3390/ijms25020838

**Published:** 2024-01-09

**Authors:** Christina Wagner, Felix F. Fuchsberger, Bernd Innthaler, Robert Pachlinger, Irene Schrenk, Martin Lemmerer, Ruth Birner-Gruenberger

**Affiliations:** 1Pharmaceutical Sciences, Baxalta Innovations (Part of Takeda), 1220 Vienna, Austria; 2Institute of Chemical Technologies and Analytics, Technical University of Vienna, 1040 Vienna, Austria

**Keywords:** adeno-associated virus vectors, nanobodies, magnetic beads, automated mass photometry, analytical ultracentrifugation

## Abstract

Mass photometry (MP) is a fast and simple analysis method for the determination of the proportions of subpopulations in an AAV sample. It is label-free and requires minimal sample volumes between 5–10 µL, which makes it a promising candidate over orthogonal techniques such as analytical ultracentrifugation (AUC), cryo-transmission electron microscopy (Cryo-TEM) or charge-detection mass spectrometry (CDMS). However, these methods are limited in their application to purified samples only. Here we developed a purification step based on single-domain monospecific antibody fragments immobilised on either a poly(styrene-divinylbenzene) resin or on magnetic beads prior to MP analysis that allows the quantification of empty, partially filled, full and overfull AAV vectors in crude cell extracts. This is aimed at identifying potentially promising harvest conditions that yield large numbers of filled AAV vectors during the early stages of the viral vector development platform, e.g., the type of transfection reagent used. Furthermore, we provide a direct comparison of the automated and manual handling of the mass photometer with respect to the quantities of AAV subspecies, molar mass of the capsid and payload, and highlight the differences between the “buffer-free” sample measurement and the “buffer-dilution” mode. In addition, we provide information on which candidates to use for calibration and demonstrate the limitations of the mass photometer with respect to the estimation of the capsid titer.

## 1. Introduction

In gene therapy, recombinant adeno-associated viruses are one of many vector platforms for the transportation of genomic cargo in patients for the treatment of rare diseases. As opposed to, e.g., retroviruses or lentiviruses, recombinant adeno-associated viruses (rAAVs) are characterised by their low immunogenicity and toxicity, which makes them attractive genomic material delivery systems [1,2,3]. There are currently seven commercially available AAV-based gene therapy products approved by the European Medicines Agency (EMA) or the US Food and Drug Administration (FDA) aimed at treating lipoprotein lipase deficiency (Glybera), retinal dystrophy (Luxturna), spinal muscular atrophy (Zolgensma), Duchenne muscular dystrophy (Elevidys), hemophilia B (Hemgenix), aromatic L-amino acid decarboxylase (AADC) deficiency (Upstaza) and haemophilia A (Roctavian), respectively [4,5,6,7,8,9,10,11]. In addition, Zynteglo, a lentivirus-based gene therapy product has been approved by the EMA and FDA for patients suffering from beta thalassemia intermedia and major [12,13].

AAV capsids are non-enveloped viruses of about 25 nm in diameter and belong to the genus *Dependoparvovirus* within the family *Parvoviridae* [14,15]. As the genus already suggests, AAVs depend on a helper virus for efficient replication, such as the herpes simplex virus (HSV) or adenovirus (Ad) [16,17,18]. They are made up of three viral proteins VP1, VP2 and VP3 differing in the length of their N-terminus. VP1, VP2 and VP3 occur in a ratio of about 1:1:10 yielding a total sum of 60 interlocking proteins which form the icosahedral capsid structure [7,19,20,21]. The packaging capacity of an AAV particle is restricted to single stranded DNA up to 4.7 kb [22]. To date, there are 12 naturally occurring serotypes, which vary in their receptor binding domains defining the tropism of the AAV vectors [1,23]. The genetic engineering of variable regions on the capsid surface allows for the manipulation of the transduction efficiency of the AAV particles and expands the possibilities of designing tailored AAV vector-based gene therapeutics for specific diseases [1,24].

As the production of AAV vectors is a complex process that is influenced by multiple factors such as cell line or plasmid ratios, not only are filled AAV particles generated but also empty and partially loaded capsids [25]. The latter represent AAV species that incorporate only parts of the intended payload and are considered neither full nor empty. In addition to the incorporation of fragmented versions of the vector genome, it is possible that more than one copy of the genomic material is inserted into the AAV capsid but is limited to the maximum loading capacity of the virus particle. This is considered an overfilled particle and, together with empty and partially filled AAV capsids, makes up the unwanted byproducts that are coproduced alongside the desired filled AAV particles. To guarantee a safe and efficacious application of a gene therapeutic, it is crucial to assess certain critical quality attributes (CQAs). One of the CQAs is the determination and quantification of impurities (empty, partially filled and overfull AAV vectors) as they may pose a risk to the recipient due to potential adverse health effects [26,27,28,29]. To date, these impurities are assessed by analytical ultracentrifugation (AUC), transmission electron microscopy (TEM) or charge-detection mass spectrometry (CDMS) [30,31,32,33]. As opposed to chromatography-based methods in combination with UV detection at 260 nm and 280 nm, AUC, TEM and CDMS are able to distinguish between AAV subspecies other than empty and filled particles, which is attributed to the high resolution of the instruments. Despite providing quantitative information on the proportions of AAV subpopulations, these methods are limited in their sample throughput and lack short turnaround times [31,34]. While CMDS and TEM demand less sample material than AUC, the latter is still the standard method for the assessment of the proportions of AAV subspecies [35]. 

Mass photometry (MP) offers a fast and simple alternative to the previously mentioned analytical techniques for the quantification of empty, partially filled, full and overfull AAV particles. With analysis times of 1–2 min, MP is significantly superior to orthogonal techniques, such as CDMS (2 h), TEM (2 h–6 h) and AUC (6 h), respectively [26]. In addition, mass photometry stands out by its low sample consumption of 5–10 µL containing 1 × 10^11^ cp mL^−1^ and minimal sample preparation, as no labelling nor immobilization steps are required [36,37]. Furthermore, the possibility of operating the instrument in an automated manner allows for a higher sample throughput than with CDMS, TEM or AUC. MP is based on the detection of reflected light from the cover slip interfering with scattered light from AAV particles attaching and detaching from the glass slide at the solid–liquid interface. The attaching and detaching of the particles are considered as binding and unbinding events, respectively. All detected binding events are summarised under the term “binding count rate”. Depending on the detected contrast, particles are registered as empty, partially filled, full or overfull AAV fractions and can subsequently be assigned a molecular mass. Hence, it is possible to confirm the molar mass of the packaged transgene in filled AAV particles, which has been described in our previous article [38].

Here we present an extended investigation of the mass photometer following up on our previously published article on the quantification of empty, partially filled and full adeno-associated virus vectors using mass photometry [38]. We provide additional information on the instrument’s performance when operated automatically and manually and compare the “buffer-dilution” mode with the “buffer-free” mode, as both can be used for sample measurement. Furthermore, we show that the variation in the number of binding events between repetitive measurements results in high deviation of the estimated capsid titer to ELISA titer, regardless of manual or automated mass photometry. In addition, we developed a method to assess AAV subspecies from crude cell material using single-domain monospecific antibody fragments immobilised on either a poly(styrene-divinylbenzene) resin or on magnetic beads followed by MP analysis. This shortcut allows for an estimation of AAV subspecies during the upstream process omitting labour-intensive cleanup steps such as diafiltration, ultracentrifugation, affinity chromatography, etc. aiming at a more efficient and less time-consuming screening of different transfection conditions, cell lines, transgenes, etc. that yield high titers of filled AAV capsids. We observed differences in the proportions of AAV subspecies when applying two different transfection reagents which are used for the introduction of plasmids into the cells. While transfection mix A yielded large quantities of partially filled and small numbers of filled AAV vectors, transfection mix B increased the titer of filled AAV capsids almost threefold according to MP. 

## 2. Results and Discussion

In our previous article we established a method for the quantification of empty, partially filled and full AAV vectors using the Samux Mass Photometer (Refeyn Ltd., Oxford, UK) [38]. Despite being a fast analytical method (1–2 min analysis times) that requires small sample volumes (2–10 µL), the manual operation of the instrument requires constant control and supervision. The number of samples per run is limited to six, which demands frequent replacement of the sample gasket and glass slides. To address this challenge, Refeyn Ltd. launched an additional robotic module (AUTO SamuxMP, Refeyn Ltd., Oxford, UK) that enables the automatic operation of the Samux MP, including sample dilution prior to measurement. The number of analysed samples was increased from six to twenty four, which allows a higher sample throughput before exchanging the sample gasket. The automated pre-dilution of a sample in a 96-well plate additionally saves time and effort and makes mass photometry a more attractive analytical technique for the quantification of the proportions of the AAV subpopulations as opposed to AUC, TEM or CDMS, which require more labour-intensive sample preparation. 

### 2.1. Selection of Calibration Samples 

Prior to sample measurement, a calibration curve is set up which assigns a measured ratiometric contrast value to a given molar mass. This allows the determination of the molar mass of each registered binding event. First, we investigated the effect of different calibration analytes of different molecular weights. Thyroglobulin (TG) was selected as the low molecular weight calibrant with a molar mass of 670 kDa. The second, high molecular weight calibrant, was a commercially available empty AAV9 (AAV9e) purchased from Progen, with a molar mass of 3.74 MDa. These analytes had been used in our previous study [38]. In addition, we investigated whether we could replace TG with the zero point. At the zero point, the molar mass and the ratiometric contrast are zero. This would save not only time but also omit the use of another calibration sample. The third option we tested was to include all three data points (zero, TG and AAV9e) for the generation of the calibration curve. As the analyte, we selected an AAV9 sample consisting of ~33% empty, ~6% partially filled, ~52% full, ~8% overfull AAV particles determined with AUC and confirmed with MP [38]. Figure 1 depicts the obtained results regarding the calculated molar masses of the empty and full AAVs, the transgene and the proportions of the AAV subpopulations. As expected, there is no significant difference in the calculated % of AAV subpopulations and molar masses of the capsids and transgene between the three calibrations models (Appendix A). The three data points-based calibration (zero, TG and AAV9e) had a coefficient of determination (R^2^) < 1 as opposed to the calibration models using two data points (TG and AAV9e; zero and AAV9e; Appendix A). For further experiments we used the zero point and AAV9e for mass calibration, as this calibration model omits the use of a second calibration standard, thus, qualifies for working in a more convenient and time-efficient manner.

### 2.2. Instrument Focusing with and without Buffer

There are two options to measure a sample on the mass photometer. The “dilution-free” mode is based on automatically adjusting the focus on the interface between the solid phase (glass slide) and the gas phase (air). The “buffer-dilution” approach uses buffer instead of air, where the buffer is pipetted into the well of the gasket, followed by the focusing step and finally adding the sample which leads to an additional dilution of the sample. A schematic illustration is given in Figure 2.

In our previously published article we used the “buffer-dilution” focusing without exception as a dilution step prior to measurement was necessary for all samples due to high capsid titers. However, according to Refeyn Ltd. both options are suitable for the generation of accurate results. Thus, we tested the “buffer-dilution” and “dilution-free” focusing modes and compared the results with respect to the quantities of AAV subpopulations, molar mass of empty/full AAVs and transgene, and binding count rates (Figure 3). Therefore, we selected the same AAV9 sample as described in Section 2.1 comprising ~33% empty, ~6% partially filled, ~52% full, ~8% overfull AAV particles. For the “buffer-free” mode the sample was pre-diluted manually. All subsequent measurements were carried out under automated instrument operation. Results show that the “buffer-free” approach performed better with respect to the calculated molar mass of the empty/full AAVs and transgene (Figure 3). In general, the variation between the samples measured using the “buffer-free” mode is higher than for the “buffer-dilution” mode which is reflected in higher standard deviations (Appendix A). The variation in the number of binding events between both modes is attributed to the lack of mixing the sample with the buffer when using the “buffer-dilution” mode. When operating the instrument manually, the sample had been mixed vigorously in the well by pipetting it up and down before measurement to ensure a homogenous distribution and adequate dilution of the AAV capsids (part of our previous publication [38]). When doing the automated “buffer-dilution”, we observed, that the robot does not mix the sample with the buffer but pipettes it onto the bottom of the sample well displacing the buffer from the bottom resulting in a higher number of binding and unbinding events than when mixing it manually. However, for the determination of quantities of AAV subspecies and molecular weight of empty/full capsids and the payload both measurement modes are suitable.

### 2.3. Manual vs. Automated Instrument Operation

As a consequence of lacking sample homogeneity when using the “buffer-dilution” mode under automatic instrument operation, we investigated and compared the performance when operating the Auto SamuxMP manually and automatically with respect to quantities of AAV subpopulations, molar mass of the capsids and transgene, and binding count rate. Therefore, an AAV9 sample (~33% empty, ~6% partially filled, ~52% full, ~8% overfull capsids) was measured five times by automated pipetting and results were compared to data obtained under manual instrument operation that had been investigated in our previous publication [38]. The comparison of both operation modes is visualised in Figure 4. Data from the automated handling is given in Appendix A. The results of manual instrument operation had been published in our previous article [38]. Figure 4 shows good comparability between both operation modes. While generally higher standard deviations were observed for measurements that had been carried out under automated operation with respect to the calculated quantities of AAV subspecies, the deviation of the calculated transgene size from the expected value was 2.5% (automated handling), which is lower than for the manual instrument operation (3.3% [38]). The highest RSD was <19% with respect to the measured proportions of the AAV subpopulations and molar mass of capsids and payload (automated handling), as opposed to a max. RSD ~10% for the manual instrument operation [38]. However, there are no significant differences between automated and manual instrument operation regarding the calculated proportions of AAV subspecies and molar masses of the capsids and genomic cargo. The variation in the number of binding events under automated instrument handling is a consequence of the omitted mixing of the sample by the robotic arm when pipetted into the sample gasket containing the buffer and had been observed when investigating the “buffer-free” and “buffer-dilution” focusing modes in Section 2.2.

### 2.4. Titer Estimation

Furthermore, we explored a feature of the SamuxMP that provides a rough estimation of the capsid titer and is based on the registration of the number of binding events. However, Refeyn Ltd. does not explicitly define, how precisely the mass photometer estimates the sample titer. Since it is not possible to generate constant binding/unbinding count rates with the MP when measuring a sample multiple times, a precise determination of the sample titer is not possible. To investigate this, we tested four AAV samples differing in their serotype (AAV8 and AAV9) and concentration (4.3 × 10^11^, 2 × 10^11^, 1.5 × 10^11^ and 1 × 10^11^ cp mL^−1^). All samples were measured in five replicates. Because we have encountered that manual instrument operation yields a smaller RSD of binding events as opposed to running the mass photometer in an automatic mode (Figure 4), we additionally tested, if the calculated titers better matched the expected ones when screening the samples manually. The results in Figure 5 confirm that under the automatic operation of the instrument, the standard deviation is generally higher than for the manual handling of the instrument. The significant fluctuation of the binding count rates between the measurements demonstrates the limitation of the instrument. When measured in manual mode, AAV8 samples yield lower capsid titers as opposed to the automatic handling of the instrument due to fewer particles attaching and detaching from the glass surface. The AAV9 samples show smaller standard deviations and a better comparison between manual and automated operation regarding AAV concentration than the AAV8 samples. In general, the calculated capsid titers (generated with MP) were lower compared to expected sample titers obtained with ELISA.

To further address this, we repeated the experiments but selected the buffer-dilution focusing option instead of the buffer-free focusing option as we found that the latter showed a greater standard deviation in the binding events than when using the buffer-dilution mode (Section 2.2). The results show an overestimation of the capsid titers when using the buffer-dilution focusing compared to the buffer-free focusing regardless of whether the instrument was operated manually or automatically and irrespective of the serotype used. When running the instrument manually, lower standard deviations were obtained than for the automated handling of the instrument. In addition, we observed a significant overestimation of the capsid titers under automated performance which is attributed to the lack of mixing the sample with the buffer. In general, the standard deviation was smaller for titers obtained with buffer-free focusing compared to buffer-dilution focusing. However, we have found no correlations so far that indicate a concentration- or serotype-dependency of the “rough titer estimation” feature but will be explored further.

### 2.5. Cleanup of AAVs from Harvest

The determination of full/empty ratios in crude AAV harvests is a very challenging task, hence, we developed a purification process which is based on the manufacturer’s protocol [39] and which provides an easy and relatively fast insight into the heterogeneity of AAV subpopulations at early stages of the development platform and could aid in adapting and optimising vector manufacturing conditions. Promising transfection conditions, e.g., the type of transfection reagent, which result in high quantities of the desired filled AAV capsids can be easily identified omitting laborious purification steps for the generation of relatively pure sample material required for MP analysis. To demonstrate this, we developed an easy purification procedure based on nanobodies immobilised on poly(styrene-divinylbenzene) beads (POROS™ CaptureSelect™ AAV9 Affinity Resin) which allows to specifically extract AAV9 capsids from crude cell material for the relative quantification of AAV species (empty, partially filled, full and overfull) using MP. A schematic illustration of the cleanup process is given in Figure 6.

After determining the best operating conditions of the instrument, we proceeded with the clean-up of AAV9 capsids from harvest material and applied our protocol to two different cell extracts which had been generated using two types of transfection reagents (transfection mix A and B). To confirm the proportion of the different AAV subpopulations after cleanup from harvest material, we performed an AUC analysis. The results in Figure 7 show that MP and AUC data agree well with one another with respect to quantities of AAV subpopulations. However, for AAV capsids generated with transfection mix A, the quantities of filled particles differed between MP (11.2%) and AUC (1.6%). Compared to MP, AUC overestimates the fraction containing empty and underestimates the quantity of filled AAV capsids. However, the findings demonstrate the great potential of mass photometry to assess AAV subpopulations in crude cell extracts and acts as fast and simple alternative to AUC.

#### Magnetic Beads

Because our cleanup protocol using POROS™ CaptureSelect™ AAV9 Affinity Resin does not have the potential to be fully automated due to centrifugation steps during the cleanup process, we used magnetic beads for the purification of AAVs from harvest samples. We developed a procedure based on the manufacturer’s manual [40] for the extraction of AAVs from crude cell material using Dynabeads™ CaptureSelect™ AAVX Magnetic Beads that has the potential to be run automatically with pipetting robots, such as Andrew+, and will be explored further in future studies but will not be discussed here. The broad range of applicable serotypes include AAV1 to AAV8, AAVrh10 and synthetic serotypes [40] due to the AAVX affinity ligand. Hence, we have observed lack of affinity against AAV9. However, Florea et al. tested the POROS™ CaptureSelect™ AAVX Affinity Resin containing the same nanobody as immobilised on the magnetic beads and were able to bind AAV9 with relatively high efficiency [41]. Because we have found no specific binding of AAVX ligands to AAV9 capsids, we tested the magnetic beads on AAV8-based cell extracts. The mass histogram in Figure 8 shows two defined fractions which can be assigned to the empty and filled AAV populations according to MP and is therefore less homogeneous than the mass histogram of AAV9. AUC measurements were omitted here, due to limitations in the sample volumes and capsid titers required for analysis. However, orthogonality between both analysis techniques was observed in previous investigations (see Section 2.5) and will be explored further in follow-up studies.

## 3. Materials and Methods

### 3.1. Sample Preparation

For the AAV cleanup from crude cell material, POROS™ CaptureSelect™ AAV9 Affinity Resin and Dynabeads™ CaptureSelect™ AAVX Magnetic Beads were purchased from ThermoFisher Scientific (Waltham, MA, USA). The affinity resin/magnetic beads were washed twice with TBS-based buffer and mixed with our in-house produced cell debris-free supernatant containing the desired AAV9/AAV8 capsids, respectively. The mixtures were incubated at room temperature for 20 min, spun down/magnetically separated, the supernatant discarded and the resin/magnetic beads washed twice with TBS-based buffer and water, respectively. The captured AAV capsids were then eluted from the resin/magnetic beads using an HCl-based buffer at pH 3 and immediately neutralised with Tris-based solution of slightly alkaline pH before analysed on the mass photometer. All in-house produced AAV samples were diluted prior to MP measurement to a final concentration between 1 × 10^11^ and 2 × 10^11^ cp mL^−1^ using HPLC-grade phosphate-buffered saline (PBS) obtained from Sigma Aldrich (Saint Louis, MO, USA). Isopropanol, necessary for cleaning the silicon sample gaskets was purchased from Merck (Darmstadt, Germany). “Empty” AAV9 capsids used for calibration was purchased from Progen (Heidelberg, Germany) and was used at a final concentration of ~3.3 × 10^11^ cp mL^−1^.

### 3.2. Measurements and Experimental Setup

AUC analysis was carried out at 15,000 rpm and 18 °C on a Beckman AUC Optima instrument (Beckman Coulter, Brea, CA, USA) equipped with an An-50 Ti analytical rotor (Beckman Coulter, Brea, CA, USA) and a total number of 150 scans per sample. The MP measurements were performed on the SamuxMP Auto (Refeyn Ltd., Oxford, UK) instrument. Compared to the standard SamuxMP, the SamuxMP Auto features automated sample handling due to an integrated robotic arm omitting constant instrument supervision and control. The 24-well sample gaskets and the glass slides were purchased from Refeyn Ltd. (Oxford, UK). A calibration was carried out prior to each measurement using the zero point as the first data point and the “empty” AAV9 vector (3.75 MDa, Progen Biotechnik GmbH, Heidelberg, Germany) as second data point. For the establishment of the calibration, “empty” AAV9 was measured at the beginning and ending of each run sequence. Since in the automated operation mode, the robot does not mix sample with the buffer provided in the well of the gasket when selecting “buffer-dilution” as measurement option, the samples were manually pre-diluted before pipetting 10 µL of the solution directly into the well of the cassette and performing the “buffer-free” focusing of the laser. Each sample was recorded in a movie for 60 s visualising the binding and unbinding events of the AAV particles which are then converted into a histogram representing the distribution of the registered molecular masses.

### 3.3. Data Collection and Processing

Measurements were recorded using AcquireMP 2.4.2 (Refeyn Ltd., Oxford, UK) and analysed with DiscoverMP (v2023 R1.2) (Refeyn Ltd., Oxford, UK). For data analysis, the bin width was set to 40 for all measurements. A Gaussian function was used to fit the ratiometric contrast distribution yielding the molecular weight of the respective subpopulation. All bar plots were visualised with Matlab R2020b (MathWorks, Natick, MA, USA).

## 4. Conclusions

Following up on our recent publication, where we demonstrated the high potential of mass photometry as opposed to AUC, TEM and CDMS for the quantification of AAV subspecies, we further investigated the instrument’s capabilities by comparing the newly featured instrument automation to manual sample handling with respect to the proportion of AAV subpopulations, molar masses of the capsid and payload, and binding count rates. In general, there are no significant differences between automated and manual instrument operation. However, we observed higher standard deviations for measurements carried out under automated operation than when handling the instrument manually with respect to the calculated quantities of AAV subspecies, the binding count rate and the rough estimation of the capsid titer. The latter demonstrates the limitation of the instrument as significant fluctuations in the registered binding events were observed. The more AAVs attach and detach from the glass surface, the higher the calculated capsid titer. The variation in the binding count rates between samples measured multiple times is reflected in a high standard deviation. If constant binding count rates between the replicates were achieved, it would be possible to gain precise information on the AAV titer. The calculation of the capsid titer using MP would have the potential to replace the more laborious ELISA in the future.

Furthermore, we compared the “buffer-dilution” mode with the “buffer-free” mode, as both can be used for sample measurement. While the first introduces an additional dilution step, the latter measures the sample in its initial state. The results showed, that under automated instrument operation and using the “buffer-dilution” option, the sample is not mixed with the buffer that had been provided in the sample well. The robot pipettes the sample onto the bottom of the well of the gasket resulting in a displacement of the buffer from the bottom which is mirrored in a higher number of binding and unbinding events than when using the “buffer-free” mode (manual pre-dilution prior to analysis). However, both analysis options are suitable for the assessment of the quantities of AAV subpopulations and the molecular weight of the incorporated genomic cargo.

After the identification of the best operating conditions, we proceeded with the cleanup of AAVs from harvest material using single-domain monospecific antibody fragments covalently attached onto the surface of poly(styrene-divinylbenzene) beads. We were able to distinguish between AAV fractions that had been generated using two types of transfection reagents (A and B) by first, extracting the AAVs from the crude cell material and then, analysing it with mass photometry and AUC. The MP and AUC data of extracted AAVs agreed well with one another demonstrating the good comparability between these orthogonal methods. To expand this further, we used nanobodies immobilised on magnetic beads for the isolation of AAVs from harvest material followed by MP analysis as these procedures have the potential to be fully automated and are paving the way for high throughput full/empty analyses, especially during the early development phase within the upstream process. During this stage, potentially promising harvest conditions that yield a high number of the desired filled AAV vectors can be easily selected and pursued further in follow-up experiments up to large-scale approaches.

## Figures and Tables

**Figure 1 ijms-25-00838-f001:**
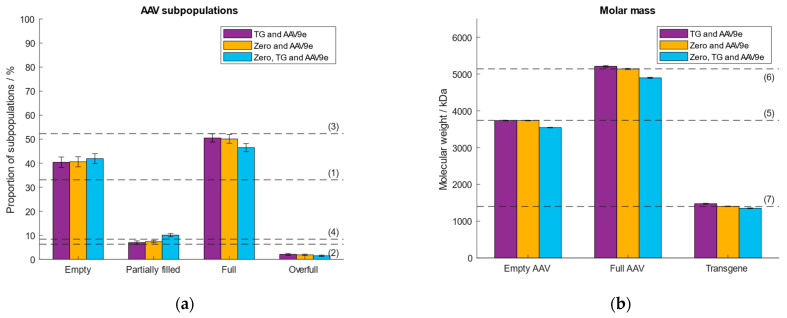
Evaluation of the performance of different calibration curves using either TG and AAV9e (purple), zero and AAV9e (orange) or zero, TG and AAV9e (blue). Dashed lines (1–7) represent (**a**) the expected proportion of the empty, partially filled, full and overfull AAV populations, (**b**) the absolute molar masses of the empty and filled AAVs capsids and the transgene, respectively.

**Figure 2 ijms-25-00838-f002:**
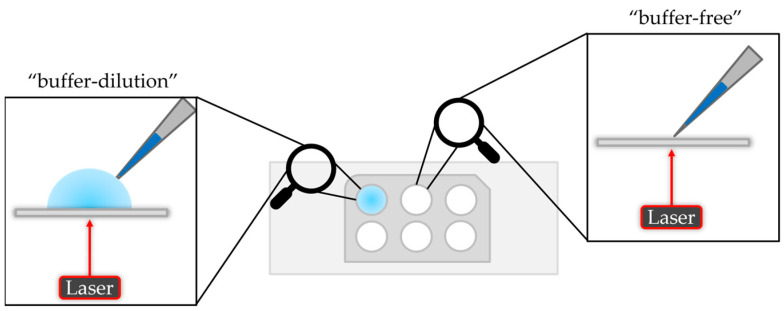
Schematic illustration of the “buffer-dilution” and “buffer-free” sample measurement.

**Figure 3 ijms-25-00838-f003:**
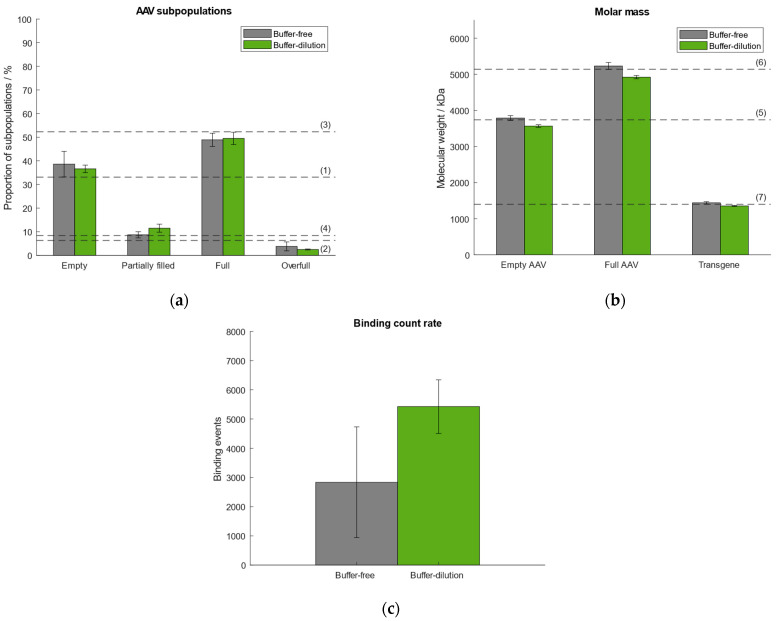
Comparison of performance using either buffer-free (grey) or buffer-dilution (green) focusing mode. Dashed lines (1–7) represent (**a**) the expected proportion of the empty, partially filled, full and overfull AAV populations, (**b**) the absolute molar masses of the empty and filled AAVs capsids and the transgene, respectively. (**c**) Comparison of binding count rates between both focusing modes.

**Figure 4 ijms-25-00838-f004:**
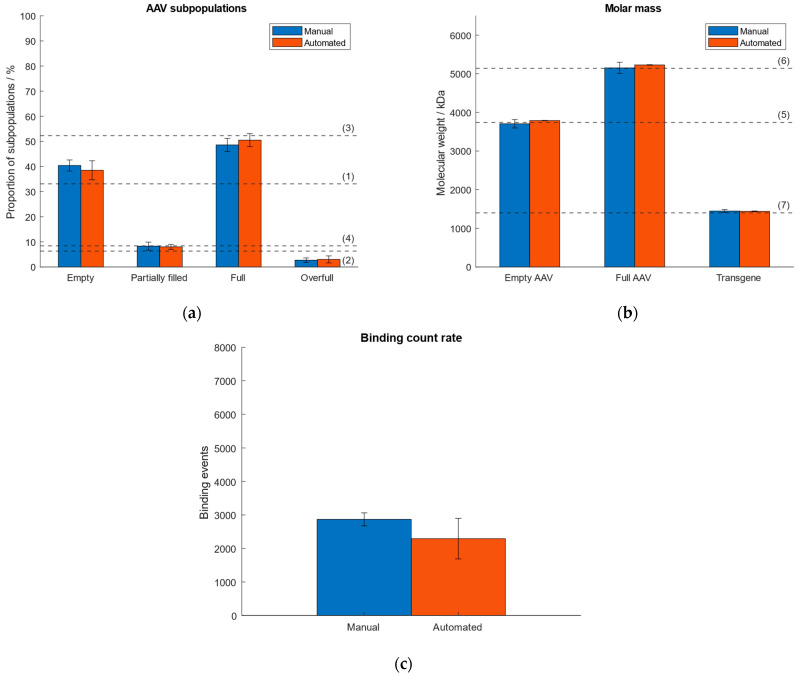
Comparison of manual and automated instrument operation. Dashed lines (1–7) represent (**a**) the expected proportion of the empty, partially filled, full and overfull AAV populations, (**b**) the absolute molar masses of the empty and filled AAVs capsids and the transgene, respectively. (**c**) Comparison of binding count rates between manual and automated operation mode.

**Figure 5 ijms-25-00838-f005:**
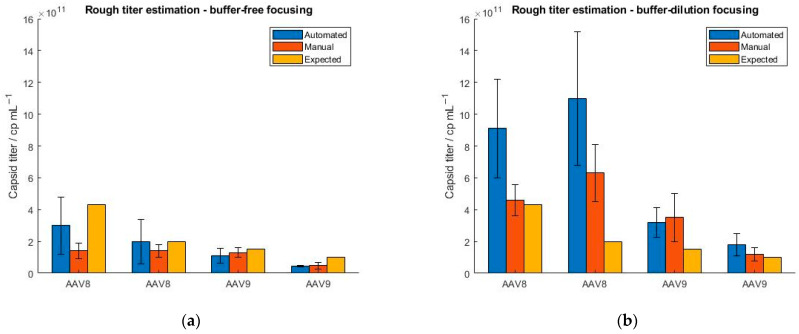
Evaluation of the “rough titer estimation” feature of the SamuxMP using two different serotypes (AAV8 and AAV9) at four different concentrations (4.3 × 10^11^, 2 × 10^11^, 1.5 × 10^11^ and 1 × 10^11^ cp mL^−1^). The samples were measured under automated (blue) and manual (orange) instrument operation and obtained capsid titers were compared to ELISA data (yellow). (**a**) buffer-free focusing, (**b**) buffer-dilution focusing mode.

**Figure 6 ijms-25-00838-f006:**
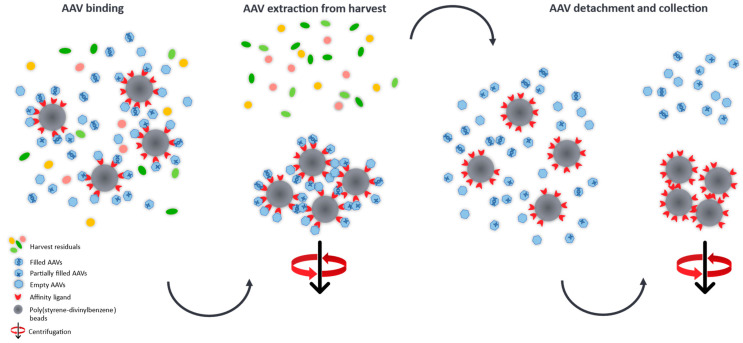
Schematic illustration of the AAV cleanup from harvest material using nanobodies covalently attached to poly(stryrene-divinylbenzene) beads. The immobilised affinity ligands selectively capture the AAVs from the crude cell extract and can then be separated from the harvest residuals by gravitational force. After elution from the binding sites, the AAVs are collected and measured by mass photometry for the quantification of AAV subpopulations.

**Figure 7 ijms-25-00838-f007:**
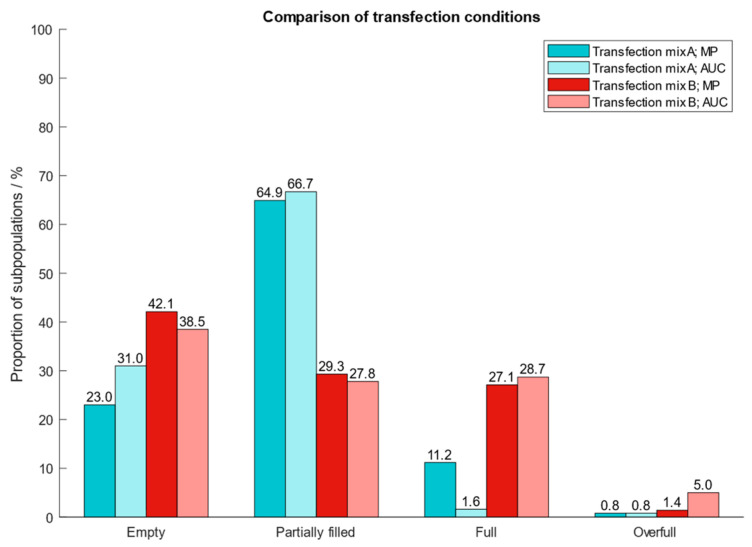
Proportion of AAV9 subspecies purified from crude cell extracts which had been produced by using two different transfection reagents for the introduction of plasmids into the cells. Blue: transfection mix A; red: transfection mix B.

**Figure 8 ijms-25-00838-f008:**
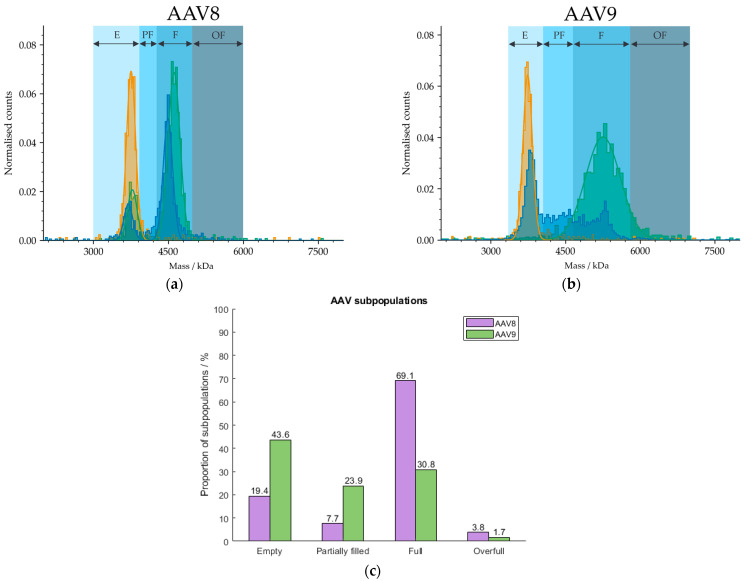
Mass histograms of (**a**) AAV8 and (**b**) AAV9. AAV8 and AAV9 capsids were extracted from curde cell extracts using Dynabeads™ CaptureSelect™ AAVX Magnetic Beads and POROS™ CaptureSelect™ AAV9 Affinity Resin, respectively. E, PF, F and OF stand for empty, partially filled, full and overfull, respectively. Samples comprising meC (orange) and mfC (green), respectively, were selected for each serotype to set the AAV limits for the analysis of an AAV sample (blue). (**c**) Barplot visualising the proportions of AAV subspecies in the AAV8 (purple) and AAV9 (green) extracts from harvest material. Results were obtained with MP.

## Data Availability

Data are contained within the article and Appendix A.

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
