# Peer review of "Automated Mass Photometry of Adeno-Associated Virus Vectors from Crude Cell Extracts"

_ijms, 2024, doi:10.3390/ijms25020838_

Round 1
Reviewer 1 Report
Comments and Suggestions for Authors
Being a fast and simple analysis method for the determination of the proportions of subpopulations in an AAV sample, as well as being label-free and requiring small volumes, MP is potentially a very promising candidate over orthogonal techniques including AUC, Cryo-TEM, or CDMS. In the paper entitled “Automated Mass Photometry of Adeno-Associated Virus Vectors from Crude Cell Extracts”, Wagner et al 1) selected calibration samples, 2) compared the differences between the “buffer-free” sample measurement and the “buffer-dilution” mode, 3) provided a direct comparison of the automated and manual handling of the mass photometer with respect to quantities of AAV subspecies, molar mass of capsid and payload, 4) demonstrated the limitation of the mass photometer with respect to estimation of the capsid titer, and 5) developed a purification step based on single-domain monospecific antibody fragments immobilized on either a poly(styrene-divinylbenzene) resin or on magnetic beads prior to MP analysis that allowed the quantification of empty, partially filled, full and overfull AAV vectors in crude cell extracts. Thorough investigation is needed to determine how useful the MP analysis is for upstream, even downstream preparation of AAV manufacture. Hence, I do have some questions about this manuscript.
1) No statistical analyses have been presented for all data shown. For example, regarding buffer-free” sample measurement and the “buffer-dilution” mode, the authors declare that buffer-free” approach performed better with respect to the calculated molar mass 172 of the empty/full AAVs and transgene. Because of no statistical analysis, it is not confident to draw a conclusion.
2) Tests and comparisons should be repeated multiple times and occasions to find out the reproducibility and rule out any random events.
3) Multiple AAV serotypes should be tested to find out the usefulness of the MP on which serotypes. The authors mainly focused on AAV9, and AAV8 on certain occasions. It would be better to include some other commonly used serotypes in their assays.
4) For the automated instrument operation, the authors declare that under the sample is not 389 mixed with the buffer that had been provided in the sample well. The robot pipettes the 390 sample onto the bottom of the well of the gasket resulting in a displacement of the buffer 391 from the bottom which is mirrored in a higher number of binding and unbinding events 392 than when using the “buffer-free” mode (manual pre-dilution prior to analysis). Is there a solution to resolve this issue? If yes, what are the results?
5) For cleanup of AAVs from harvest material using beads, the authors should do a better job to establish and present a protocol or protocols which can be used on different serotype and settings. For example, how well do the beads perform with different subpopulations in the crude extracts?
6) What is the range of AAV concentrations which can be reliably measured by the MP analysis? If the MP has the limitation to determine the titer, is that because of AAV aggregation or something else?
In summary, there are so many questions remaining regarding the MP analysis. I hope the authors can at least address some of the important questions before we can accept the manuscript for publication.
Comments on the Quality of English LanguageMinor editing of English language required
Reviewer 2 Report
Comments and Suggestions for Authors
Photometry is a new alternative technique for AAV characterization. This paper comprehensively validated this method including newly developed automated workflow. Although scientific novelty is not very obvious, it still can be a good reference for researchers working in AAV field. It can be considered for acceptance after addressing the following questions.
1) There are more approved AAV gene therapy than listed in Introduction, including Upstaza and Roctavian.
2) There are many reference sources not found. Either format is wrong, or reference doesn't exist.
3) What does Binding count rate mean exactly in the context of this technique? Better to explain in detail.
4) For calibration curve, were only 2 data points used? How to ensure its accuracy?
5) Better to show the calibration curves in the paper.
Round 2
Reviewer 1 Report
Comments and Suggestions for Authors
I appreciate the authors' attention to the comments in my previous review. The authors have addressed/clarified my previous concerns.
Reviewer 2 Report
Comments and Suggestions for Authors
The revised version can be accepted for publication.